# Prognostic Significance of Initial Absolute Lymphocyte Count in Adjuvant Radiotherapy for Pancreatic Adenocarcinoma

**DOI:** 10.3390/biomedicines10092190

**Published:** 2022-09-05

**Authors:** Jaesung Heo, O Kyu Noh

**Affiliations:** 1Department of Radiation Oncology, Ajou University School of Medicine, Suwon 16499, Korea; 2Department of Biomedical Informatics, Ajou University School of Medicine, Suwon 16499, Korea; 3Office of Biostatistics, Ajou Research Institute for Innovative Medicine, Suwon 16499, Korea

**Keywords:** pancreatic cancer, blood lymphocyte count, adjuvant radiation therapy, prognostic factor

## Abstract

Background: This study aimed to investigate the impact of absolute lymphocyte count (ALC) on clinical outcomes in patients treated with adjuvant RT with or without chemotherapy for pancreatic adenocarcinoma. Methods: From 2001 to 2015, 68 patients underwent curative surgery followed by adjuvant RT. Chemotherapy was administered concurrently or sequentially with RT. We analyzed the clinical impact of the initial ALC level on locoregional recurrence-free survival (LRRFS), distant metastasis-free survival (DMFS), and overall survival (OS). Results: With a median follow-up of 13.7 months (range: 3.1–61.3), the 3 year OS, LRRFS, and DMFS are 25.4%, 40.0%, and 26.6%, respectively. The OS and LRRFS of the high initial ALC group (≥ 1540 × 10^6^/L) are significantly higher than that of the group with lower initial ALC (3 year OS: 32.6% vs. 18.6%, *p* = 0.036; 3 year LRRFS: 53.5% vs. 27.0%, *p* = 0.031). In multivariable analyses, initial ALC level is the significant prognostic factor affecting LRRFS (HR = 0.457, *p* = 0.028) and OS (HR = 0.473, *p* = 0.026). Conclusions: Initial ALC could have potential prognostic significance in patients with pancreatic adenocarcinoma receiving adjuvant RT with or without chemotherapy. Further studies are warranted to investigate the role of adjuvant RT, considering the initial ALC.

## 1. Introduction

Patients with pancreatic adenocarcinoma have a poor prognosis [1]. Surgery is the only treatment option for curing patients and long-term survival, and it is feasible only in 10–20% of cases at the time of diagnosis [2]. Even in the patients who received curative resection, the prognosis is poor due to the high failure rate [3,4], which necessitates the use of adjuvant therapies such as chemotherapy, radiation therapy, and combined approaches.

The benefit of adjuvant chemotherapy for pancreatic cancer is well-defined [5,6,7], but the role of adjuvant radiotherapy (RT) is controversial. In the US, adjuvant chemoradiotherapy (chemo-RT) is preferred based on the Gastrointestinal Tumor Study Group (GITSG) trial [8]. Two large retrospective studies suggested a survival benefit of adjuvant chemo-RT [9,10]. However, randomized trials performed in Europe failed to show a benefit of adjuvant chemo-RT [11], and even reported deleterious effects on survival [12]. Moreover, a network meta-analysis shows that chemo-RT plus chemotherapy is less effective in prolonging survival [13]. These conflicting results suggest a potential adverse impact of adjuvant RT on controlling the locoregional disease.

Recently, it was reported that radiation-induced lymphopenia (RIL) could negatively affect the survival of patients with pancreatic adenocarcinoma [14,15]. We hypothesized that low initial absolute lymphocyte count (ALC) could be vulnerable to the development of RIL, and it can be an adverse prognostic factor in patients receiving adjuvant RT. Our institution has a cohort of pancreatic adenocarcinoma treated with upfront adjuvant RT before using upfront adjuvant chemotherapy. Using the historical adjuvant RT cohort, we aim to investigate the clinical impact of initial ALC on overall survival (OS), locoregional recurrence-free survival (LRRFS), and distant metastasis-free survival (DMFS) in adjuvant RT for patients with resected pancreatic adenocarcinoma.

## 2. Materials and Methods

Between July 2001 and August 2015, we identified 68 patients with pancreatic adenocarcinoma who underwent curative-intent resection and upfront adjuvant RT with or without chemotherapy in our institution. According to our institutional protocol, adjuvant RT with or without chemotherapy was administered in patients with pathological stage T3 or N1 or with positive surgical resection margin. Adjuvant RT began 4 to 8 weeks after surgery, and 3-dimensional conformal radiation therapy was implemented with a dose of 45–54 Gy (1.8–2.0 Gy/day) to the tumor bed and regional lymphatics. The initial ALC value was defined as the median value among ALC values obtained within one month before the surgery date. To investigate the clinical value of initial ALC, we divided the patients into two groups, according to the median value of the initial ALC values (Figure 1).

We analyzed clinical parameters influencing LRRFS, DMFS, and OS according to the initial ALC level. The comparison between the two groups was performed with the chi-squared test or Fisher exact test for categorical variables, and the Student’s *t*-test or Kruskal–Wallis rank sum test for continuous variables. Survival outcomes were calculated by the Kaplan–Meier method. Log-rank test and Cox proportional hazard regression model were used for univariable and multivariable analyses, respectively. The variables with *p* values of less than 0.20 in univariable analysis were selected for multivariable analysis. All analyses were performed using the R statistical packages [16]. This study was performed in compliance with the Helsinki II Declaration, and was reviewed and approved by the institutional review board of Ajou University Hospital (IRB No. AJIRB-MED-2015-163). Acquisition of written informed consent was exempted.

## 3. Results

### 3.1. Patient Characteristics between the High and Low Initial ALC Groups

The initial ALC of all patients ranges from 618.6 to 3151.0 × 10^6^/L (median, 1540.0). Between the groups with high and low initial ALC (≥1540.0 × 10^6^/L vs. <1540.0 × 10^6^/L), there are no significant differences in clinical variables, except the use of chemotherapy (*p* = 0.026) (Table 1).

### 3.2. Changes of ALC over Time

The trend line of ALC shows a decrease and a recovery after surgery, and then a decrease again with the initiation of adjuvant RT (Figure 2A). The decreased ALC does not recover up to the pre-RT level, and is sustained after the end of RT. The different levels of ALC between the two groups with high and low initial ALC are maintained after the end of RT (Figure 2B).

### 3.3. Comparison of Survivals between the High and Low Initial ALC Groups

The median follow-up period ranges from 3.1 to 61.3 months (median: 13.7), and the 3 year OS, LRRFS, and DMFS are 25.4%, 40.0%, and 26.6%, respectively. The OS and LRRFS of the group with high initial ALC are significantly superior to those of the group with low ALC (3 year OS: 32.6% vs. 18.6%, *p* = 0.036; 3 year LRRFS: 53.6% vs. 27.0%, *p* = 0.031) (Figure 3A,B). There is no significant difference in DMFS between the groups (*p* = 0.376) (Figure 3C).

### 3.4. Univariable and Multivariable Analyses Affecting Survivals

Univariable analyses for clinical factors affecting survival are presented in Table 2. Due to the significant difference in chemotherapy use between the high and low ALC groups, we added this variable into the multivariable models. In multivariable analyses, initial ALC is the significant prognostic factor affecting LRRFS (hazard ratio [HR] = 0.457, *p* = 0.044) and OS (HR = 0.473, *p* = 0.026) (Table 3).

## 4. Discussion

We hypothesized that low initial ALC is associated with RIL after adjuvant RT, and both the initial ALC level and the RIL have prognostic significance. This study shows that the low initial ALC group have poor survival outcomes compared to the high initial ALC group. The LRRFS of the low initial ALC group is significantly decreased compared to that of the high initial ALC group, and the significance of LRRFS is sustained in OS (Figure 3) (Table 2). However, a decrease in the ALC level after adjuvant RT seems not to be associated with the initial ALC level (Figure 2). From day 25 to 30 after the start of adjuvant RT, the ALC level decreases similarly, regardless of the initial ALC group. It suggests that adjuvant RT dramatically impacts the level of ALC during its administration. Balmanoukian et al. report that the treatment-induced lymphopenia two months after adjuvant chemo-RT is associated with overall survival [14]. They defined treatment-related lymphopenia as less than 500 × 10^6^/L. However, we could not analyze the survival according to their definition of treatment-related lymphopenia. In most patients, the ALC levels two months after the start of RT are above 500 × 10^6^/L, and we observe that the level of ALC is variable even within one patient after adjuvant RT (Figure 2B). Our results suggest that the initial ALC level can be a significant prognostic factor affecting clinical outcomes. Previously, Clark et al. reported that the preoperative ALC was a prognostic factor in patients with pancreatic cancer receiving surgery [17]. Although they analyzed the patients treated without adjuvant therapies, the results support our prognostic significance of the initial ALC in the setting of adjuvant RT.

Patient characteristics are not different between high and low initial ALC groups, except for the administration of chemotherapy (Table 1). The high initial ALC group received more chemotherapy than the low ALC group, and this difference could affect survival outcomes. However, the level of initial ALC maintained its prognostic significance after adjusting the chemotherapy effect in multivariable analysis (Table 2). Moreover, DMFS is not significantly different between the high and low ALC groups (Figure 3C). The difference in LRRFS seems to result in a significant difference in OS between the high and low ALC groups (Figure 3A,B). The primary aim of adjuvant RT is to improve locoregional control, leading to survival benefits. However, the adjuvant RT in the low ALC group appears ineffective compared to the high ALC group. This finding suggests that the low ALC level compromises the effect of adjuvant RT. A histopathological study shows that T-cell lymphocyte infiltration in proximity to pancreatic cancer cells correlates with increased overall survival [18]. Considering circulating lymphocytes can be a pool for infiltrating lymphocytes in residual tumors, competent ALC levels can be associated with increased immunological cell death and survival [19,20]. As shown in Figure 2, adjuvant RT reduces the circulating lymphocytes count to a similar level, regardless of the high or low initial ALC group. However, the recovery of ALC over time depends on the level of initial ALC. Although it does not recover to the initial ALC level, the high initial ALC group is restored more than the low ALC group. This recovery may contribute to the immunological anti-tumor effect of the locoregional disease. Lee et al. report that the recovery of lymphocytes after definitive chemo-RT is associated with superior OS and disease-free survival [21]. They also show that the initial ALC level is associated with the recovery from acute severe lymphopenia, which is consistent with our recovery pattern of ALC after adjuvant RT.

Irradiated target volume and dose fractionation can be associated with treatment-related lymphopenia and affect the level of lymphocyte recovery [21,22,23]. We could not analyze these effects on the ALC level in this study because of the homogeneous study population using similar volume and dose fractionation. Advanced techniques or altered fractionations can be applied to save the lymphocytes level, particularly for the low initial ALC group [24,25]. However, even though these prevent a severe decrease in the ALC level, they may not be related to the ALC recovery. The dynamics and clinical impact of the ALC level should be investigated among different dose–volume fractionations and techniques.

This study does not compare the results with and without adjuvant RT. Therefore, the results of this study do not imply the benefit of adjuvant RT, even in the high initial ALC group. However, the inclusion of patients having low initial ALC levels could be one of the possible reasons for conflicting data regarding the role of adjuvant RT [8,9,10,11,12,13]. Further studies are warranted to define the role of initial ALC level in patients treated with adjuvant therapies combining RT.

The results of this retrospective study are limited by its small study population over a relatively long period, with a short follow-up time. We analyzed the selected group of patients receiving surgical resection followed by adjuvant radiotherapy, and an initial low ALC may be correlated with the occurrence of radiation-induced lymphopenia. Therefore, this study’s prognostic significance of initial ALC has a limited value. Further studies should be conducted to explore the prognostic value of initial ALC in the whole cohort receiving curative intent surgery followed by adjuvant chemotherapy with or without radiotherapy. We focused on the initial ALC level to investigate the association with adjuvant RT. However, the ALC level is dynamic over time. The results can be biased by the factors affecting the ALC level, such as combined therapies, physical and psychological stress, and medication. Therefore, the clinical impact of the post-treatment ALC level at other times should be investigated in the prospective setting. The lack of data for lymphocyte subpopulations also limits a detailed interpretation of this study.

## 5. Conclusions

Initial ALC could have potential prognostic significance in patients with pancreatic adenocarcinoma receiving adjuvant RT with or without chemotherapy. Further studies are warranted to investigate the role of adjuvant RT considering the initial ALC level.

## Figures and Tables

**Figure 1 biomedicines-10-02190-f001:**
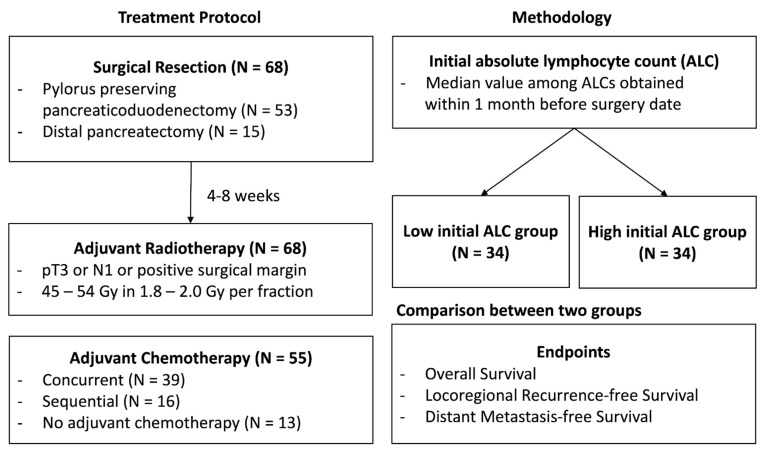
Study treatment protocol and methodology.

**Figure 2 biomedicines-10-02190-f002:**
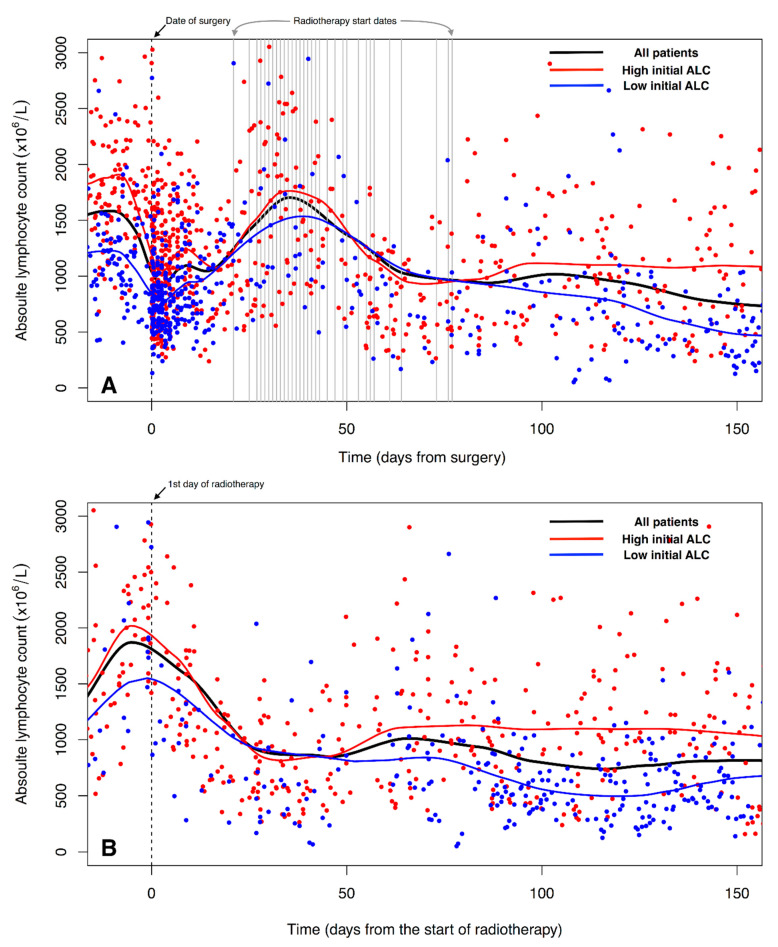
The trend of absolute lymphocyte count (ALC) of 68 patients over time. The solid lines indicate the trend of local regression. (**A**) Trend of ALC from surgery. (**B**) Trend of ALC from the start of radiotherapy.

**Figure 3 biomedicines-10-02190-f003:**
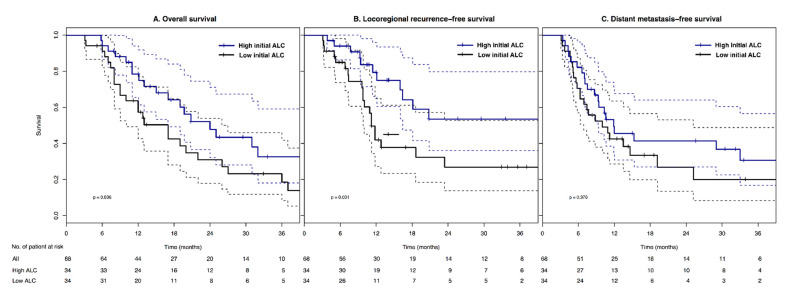
Kaplan–Meier survival curves according to the level of initial absolute lymphocyte count. Dashed lines indicate the 95% confidence interval for survival curves (solid lines). (**A**) Overall survival. (**B**) Locoregional recurrence-free survival. (**C**) Distant metastasis-free survival.

**Table 1 biomedicines-10-02190-t001:** Patient characteristics between groups with low and high initial lymphocyte count.

	Low Initial ALC (N = 34)	High Initial ALC (N = 34)	*p*-Value
Age (year)			0.556
Mean ± SD	59.8 ± 9.9	58.4 ± 10.6	
Gender			0.615
Male	20	23	
Female	14	11	
ECOG PS			0.575
0–1	27	24	
2	7	10	
Preoperative CA 19-9 (U/mL)			0.980
Mean ± SD	281.1 ± 583.9	277.5 ± 562.1	
Initial ALC (×10^6^/L)			
Mean ± SD	1217.8 ± 245.3	2049.3 ± 433.2	<0.001
Type of surgery			0.079
DP	4	11	
PPPD	30	23	
T stage			0.261
T1	0	1	
T2	0	1	
T3	32	32	
T4	2	0	
N stage			1.000
N0	11	12	
N1	23	22	
Resection margin			1.000
Positive	9	9	
Negative	25	25	
Chemotherapy			0.026
No chemotherapy	9	4	
Sequential	11	5	
Concurrent	14	25	
Radiotherapy dose (Gy)			0.245
Mean ± SD	49.9 ± 2.8	50.8 ± 3.6	

Abbreviations: ALC = absolute lymphocyte count; SD = standard deviation; ECOG PS = Eastern Cooperative Oncology Group performance status; CA 19-9 = carbohydrate antigen 19-9; DP = distal pancreatectomy; PPPD = pylorus-preserving pancreaticoduodenectomy.

**Table 2 biomedicines-10-02190-t002:** Univariable analyses for clinical variables affecting survivals.

Variable	3 Year LRRFS (%)	*p*-Value (log-Rank)	3 Year DMFS (%)	*p*-Value (log-Rank)	3 Year OS (%)	*p*-Value (log-Rank)
Age (<61 vs. ≥61)	41.2 vs. 37.9	0.932	32.0 vs. 21.0	0.378	33.3 vs. 15.2	0.138
Gender (male vs. female)	42.3 vs. 38.1	0.604	32.3 vs. 17.0	0.548	23.9 vs. 27.0	0.773
ECOG PS (0–1 vs. 2)	45.3 vs. 30.9	0.319	22.8 vs. 30.2	0.134	27.6 vs. 19.3	0.667
Preoperative CA 19-9 (low vs. high)	39.3 vs. 35.8	0.638	28.8 vs. 30.1	0.815	27.6 vs. 21.2	0.503
Initial ALC (low vs. high)	27.0 vs. 53.6	0.031	20.1 vs. 30.8	0.376	18.6 vs. 32.6	0.036
Surgery (DP vs. PPPD)	66.9 vs. 33.1	0.020	9.4 vs. 34.4	0.472	35.3 vs. 23.2	0.395
N stage (N0 vs. N1)	41.9 vs. 37.6	0.931	44.0 vs. 13.6	0.062	46.3 vs. 14.9	0.007
Resection margin (− vs. +)	48.3 vs. 12.9	0.090	28.3 vs. 27.2	0.352	32.3 vs. 0.0	0.076
Chemotherapy (no vs. sequential vs. concurrent)	34.4 vs. 33.7 vs. 46.0	0.733	20.5 vs. 35.2 vs. 25.5	0.752	17.6 vs. 23.0 vs. 30.4	0.860
Radiotherapy dose (<50.4 Gy vs. ≥50.4 Gy)	34.4 vs. 44.9	0.414	26.0 vs. 27.3	0.379	15.9 vs. 33.8	0.243

Abbreviations: LRRFS = locoregional recurrence-free survival; DMFS = distant metastasis-free survival; OS = overall survival; ECOG PS = Eastern Cooperative Oncology Group performance status; CA 19-9 = carbohydrate antigen 19-9; ALC = absolute lymphocyte count; DP = distal pancreatectomy; PPPD = pylorus-preserving pancreaticoduodenectomy.

**Table 3 biomedicines-10-02190-t003:** Multivariable analyses for clinical variables affecting survivals.

Variable	LRRFS			DMFS			OS		
	HR	95% CI	*p*-Value	HR	95% CI	*p*-Value	HR	95% CI	*p*-Value
Age (<61 vs. ≥61)							0.928	0.471–1.828	0.829
ECOG PS (0–1 vs. 2)				0.475	0.206–1.094	0.080			
Initial ALC (low vs. high)	0.457	0.214–0.978	0.044				0.473	0.244–0.916	0.026
Surgery (DP vs. PPPD)	3.765	1.100–12.884	0.035						
N stage (N0 vs. N1)				2.134	1.045–4.360	0.038	2.403	1.104–5.228	0.027
Resection margin (− vs. +)	2.682	1.110–6.483	0.028				1.909	0.898–4.058	0.093
Chemotherapy (no vs. sequential)	0.596	0.185–1.913	0.384	0.871	0.340–2.230	0.773	0.795	0.315–2.010	0.628
Chemotherapy (no vs. concurrent)	1.131	0.447–2.862	0.795	0.781	0.361–1.692	0.531	0.972	0.433–2.181	0.945

Abbreviations: LRRFS = locoregional recurrence-free survival; DMFS = distant metastasis-free survival; OS = overall survival; HR = hazard ratio; CI = confidence interval; ECOG PS = Eastern Cooperative Oncology Group performance status; ALC = absolute lymphocyte count; DP = distal pancreatectomy; PPPD = pylorus-preserving pancreaticoduodenectomy.

## Data Availability

The data presented in this study are available on request from the corresponding author. The data are not publicly available due to private information of patients.

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
