# Peer review of "Prognostic Significance of Initial Absolute Lymphocyte Count in Adjuvant Radiotherapy for Pancreatic Adenocarcinoma"

_biomedicines, 2022, doi:10.3390/biomedicines10092190_

Round 1
Reviewer 1 Report
Heo et al described the Prognostic significance of initial absolute lymphocyte count in adjuvant radiotherapy for pancreatic adenocarcinoma using retrospective samples.
The data is well presented in the figures, robust and mostly supports the authors interpretation. However, there are some major concerns/suggestions.
1. Several citations are too old.
2. Since the population size of the study is small and retrospective, the conclusion part in the abstract should be reframed.
3. Author should provide a schematic presentation of the methodology.
4. The result section should be elaborated according to the fig, table etc, (sub headings can be used)
Author Response
Responses to the Reviewer #1’s Comments
Heo et al described the Prognostic significance of initial absolute lymphocyte count in adjuvant radiotherapy for pancreatic adenocarcinoma using retrospective samples.
The data is well presented in the figures, robust and mostly supports the authors interpretation. However, there are some major concerns/suggestions.
- Several citations are too old.
=> Response: As you commented, we have updated the old citations.
- Since the population size of the study is small and retrospective, the conclusion part in the abstract should be reframed.
=> Response: We agree with your opinion. Considering this study's small size and retrospective nature, we cannot conclude that the initial ALC was a prognostic factor. We rephrased our conclusion that the initial ALC could have potential prognostic significance.
- Author should provide a schematic presentation of the methodology.
=> Response: As you suggested, we added a figure presenting methodology (Figure 1).
- The result section should be elaborated according to the fig, table etc, (sub headings can be used)
=> Response: We elaborated the results section using subheadings as you suggested.
Reviewer 2 Report
The paper nicely investigates the potential prognostic role of absolute lymphocyte count (ALC) in patients with curative intent surgery for PDAC followed by adjuvant radiotherapy. The authors show that patients with high initial ALC have significantly better overall and loco-regional recurrence-free survivals than those with low initial ALC. These findings are particularly interesting because, in the low initial ALC group of patients, a statistically significantly higher proportion of patients received adjuvant chemotherapy compared to the high initial ALC.
Although the topic would be of interest to the journal readers, however few concerns should be raised:
Major concerns:
The study cohort includes a limited number of patients during a relatively long period, with a short median follow-up time. Furthermore, it includes a selected group of patients resected for PDAC (T3 or N1 or positive resection margins). Therefore, the present analyses' results would have a limited value. These aspects should be stated as study limitations. It would be interesting to explore the same potential prognostic significance of ALC in the group of patients with curative intent surgery for PDAC with adjuvant therapies (chemotherapy and/ or radiotherapy) to see if similar results are obtained.
The multivariate analyses should include only those factors with p values less than 0.05 in univariate analyses.
It is unclear if an initial low ALC is correlated with the occurrence of radiation-induced lymphopenia, a proven factor of poor overall survival in patients resected for PDAC with adjuvant radiotherapy. An analysis of the whole cohort of patients with curative intent surgery for PDAC with adjuvant therapies (chemotherapy and/ or radiotherapy) would help answer this question.
In the Discussion, it would be better to discuss any potential explanation for the results of the present study. Why is ALC correlated with survival? A potential explanation?
Minor concerns:
In Figure 2, please show on figures which line is for low ALC and which is for high ALC. Furthermore, what are the other lines for? There should be only two lines (for low and high ALC patients). Please clarify.
In Table 1, it is unclear at Resection margin what it means yes or no (positive resection margins or negative resection margins?). Please clarify.
Author Response
Responses to the Reviewer#2’s Comments
The paper nicely investigates the potential prognostic role of absolute lymphocyte count (ALC) in patients with curative intent surgery for PDAC followed by adjuvant radiotherapy. The authors show that patients with high initial ALC have significantly better overall and loco-regional recurrence-free survivals than those with low initial ALC. These findings are particularly interesting because, in the low initial ALC group of patients, a statistically significantly higher proportion of patients received adjuvant chemotherapy compared to the high initial ALC.
=> Response: We appreciate your comments on our manuscript.
Although the topic would be of interest to the journal readers, however few concerns should be raised:
Major concerns:
#1. The study cohort includes a limited number of patients during a relatively long period, with a short median follow-up time. Furthermore, it includes a selected group of patients resected for PDAC (T3 or N1 or positive resection margins). Therefore, the present analyses' results would have a limited value. These aspects should be stated as study limitations. It would be interesting to explore the same potential prognostic significance of ALC in the group of patients with curative intent surgery for PDAC with adjuvant therapies (chemotherapy and/ or radiotherapy) to see if similar results are obtained.
=> Response: We agree with your opinion. As you commented, we have updated the study limitations.
[Limitations description in the discussion section]
“The results of this retrospective study are limited by its small study population during a relatively long period, with a short follow-up time. We analyzed the selected group of patients receiving surgical resection followed by adjuvant radiotherapy, and an initial low ALC may be correlated with the occurrence of radiation-induced lymphopenia. Therefore, this study's prognostic significance of initial ALC would have a limited value. Further studies should be conducted to explore the prognostic value of initial ALC in the whole cohort receiving curative intent surgery followed by adjuvant chemotherapy with or without radiotherapy.”
#2. The multivariate analyses should include only those factors with p values less than 0.05 in univariate analyses.
=> Response: Less than 0.05 of p-value can be one of the selection criteria for multivariable analysis. However, the selection criteria can be modified by the investigators' discretion and the factors' clinical implications. We described our selection criteria of clinical variables for multivariable analysis in the materials and method section.
[Materials and methods section]
“The variables with P values of less than 0.20 in univariable analysis were selected for multivariable analysis.”
#3. It is unclear if an initial low ALC is correlated with the occurrence of radiation-induced lymphopenia, a proven factor of poor overall survival in patients resected for PDAC with adjuvant radiotherapy. An analysis of the whole cohort of patients with curative intent surgery for PDAC with adjuvant therapies (chemotherapy and/ or radiotherapy) would help answer this question.
=> Response: We agree with your opinion. As you commented, we described the limited value of our study in the discussion section (Please refer to the authors’ response to the reviewer’s comment #1).
#4. In the Discussion, it would be better to discuss any potential explanation for the results of the present study. Why is ALC correlated with survival? A potential explanation?
=> Response: As you suggested, we discussed the potential explanation for why ALC level is associated with survival.
[Discussion section]
“The primary aim of adjuvant RT is to improve locoregional control, leading to survival benefits. However, the adjuvant RT in the low ALC group appeared ineffective compared to the high ALC group. This finding suggests that the low ALC level compromises the effect of adjuvant RT. Histopathological study showed that T-cell lymphocyte infiltration in proximity to pancreatic cancer cells correlates with increased overall survival [18]. Considering circulating lymphocytes can be a pool for infiltrating lymphocytes in residual tumors, competent ALC levels can be associated with increased immunological cell death and survival [19,20].”
[References]
- Carstens, J.L.; Correa de Sampaio, P.; Yang, D.; Barua, S.; Wang, H.; Rao, A.; Allison, J.P.; LeBleu, V.S.; Kalluri, R. Spatial computation of intratumoral T cells correlates with survival of patients with pancreatic cancer. Nat Commun 2017, 8, 15095, doi:10.1038/ncomms15095.
- Menetrier-Caux, C.; Ray-Coquard, I.; Blay, J.Y.; Caux, C. Lymphopenia in Cancer Patients and its Effects on Response to Immunotherapy: an opportunity for combination with Cytokines? J Immunother Cancer 2019, 7, 85, doi:10.1186/s40425-019-0549-5.
- Diehl, A.; Yarchoan, M.; Hopkins, A.; Jaffee, E.; Grossman, S.A. Relationships between lymphocyte counts and treatment-related toxicities and clinical responses in patients with solid tumors treated with PD-1 checkpoint inhibitors. Oncotarget 2017, 8, 114268-114280, doi:10.18632/oncotarget.23217.
Minor concerns:
#5. In Figure 2, please show on figures which line is for low ALC and which is for high ALC. Furthermore, what are the other lines for? There should be only two lines (for low and high ALC patients). Please clarify.
=> Response: We added the description for the lines in the figure and its legend.
#6. In Table 1, it is unclear at Resection margin what it means yes or no (positive resection margins or negative resection margins?). Please clarify.
=> Response: As you commented, we clarified the margin status in the table.
Round 2
Reviewer 2 Report
The authors did not adequately address all the concerns raised by the reviewers. Thus, the paper can not be accepted for publication in its current form.